# Clinical Evaluation of Underwater Discharge Plasma as a Root Canal Irrigant: A Randomized Pilot Study on Efficacy and Safety

**DOI:** 10.3390/biomedicines13102343

**Published:** 2025-09-25

**Authors:** Jeong-Hyo Lyu, Young-Hee Kim, Hyun-Sook Chung, Sang-Yoon Park, Sang-Min Yi, Soo-Hwan Byun, Sung-Woon On, Jae-Seo Lee, Byoung-Eun Yang

**Affiliations:** 1Division of Conservative Dentistry, Hallym University Sacred Heart Hospital, Anyang 14066, Republic of Korea; lyujh0519@naver.com (J.-H.L.); dnendo@gmail.com (H.-S.C.); 2Department of Artificial Intelligence and Robotics in Dentistry, Graduate School of Clinical Dentistry, Hallym University, Chuncheon 24252, Republic of Korea; kcallas2@gmail.com (Y.-H.K.); psypjy0112@naver.com (S.-Y.P.); queen21c@hallym.or.kr (S.-M.Y.); purheit@daum.net (S.-H.B.); drummer0908@hanmail.net (S.-W.O.); 3Institute of Clinical Dentistry, Hallym University, Chuncheon 24252, Republic of Korea; 4Dental Artificial Intelligence and Robotics R&D Center, Hallym University Medical Center, Anyang 14066, Republic of Korea; 5Division of Image Science in Dentistry, Hallym University Sacred Heart Hospital, Anyang 14066, Republic of Korea; 6Division of Oral and Maxillofacial Surgery, Hallym University Sacred Heart Hospital, Anyang 14066, Republic of Korea; 7Department of Oral and Maxillofacial Radiology, School of Dentistry, Chonnam National University, Gwangju 61186, Republic of Korea; jsyi16@jnu.ac.kr

**Keywords:** underwater discharge plasma, non-thermal plasma, endodontics, root canal, hydroxyl radical

## Abstract

**Background/Objectives:** Root canal therapy (RCT) aims to eliminate intracanal infection and promote periapical healing through mechanical instrumentation and chemical disinfection. Conventional irrigants, such as sodium hypochlorite (NaOCl), are effective but may exhibit limited penetration into anatomically complex root canal systems and carry the risks of cytotoxicity if extruded beyond the apical foramen or into surrounding periodontal tissues. In this pilot study, we evaluated the clinical effectiveness and safety of underwater discharge plasma (UDP) as a biocompatible alternative to NaOCl for root canal irrigation. **Methods:** A prospective, randomized clinical trial was conducted involving 30 patients who required root canal treatment. Patients were randomly allocated to the UDP (n = 15) or NaOCl (n = 15) group. All treatments were performed by a single operator following standardized protocols. Pain was assessed using the visual analog scale (VAS), and periapical healing was evaluated using the Periapical Index (PAI) at baseline, 2 months, and 4 months. Statistical analyses included the Friedman test, Mann–Whitney U test, and Fisher’s exact test. Interobserver agreement for radiographic readings was evaluated using quadratic-weighted Cohen’s kappa coefficient. **Results:** A total of 28 patients completed the study. VAS scores significantly decreased over time in both groups (*p* < 0.05), with no significant difference between the groups at any time point (*p* > 0.05). At 4 months, radiographic healing was observed in 71.4% and 92.9% of patients in the UDP and NaOCl groups, respectively (*p* > 0.05). PAI score changes and clinical success rates were comparable between groups. No adverse effects or thermal damage was reported when using UDP. **Conclusions:** UDP demonstrated short-term clinical efficacy and safety comparable to that of NaOCl. Thus, UDP may serve as a biocompatible alternative for root canal disinfection. Further large-scale and long-term studies are warranted to confirm its clinical utility.

## 1. Introduction

Root canal treatment (RCT) is performed to eliminate infected pulp tissue and prevent reinfection, thereby facilitating the healing of periapical tissues [1]. Mechanical instrumentation and chemical irrigation are complementary procedures essential to the success of RCT. Nickel–titanium (NiTi) rotary instruments and sodium hypochlorite (NaOCl) have long served as fundamental tools in root canal disinfections [2,3]. Advances in NiTi metallurgy have enhanced the flexibility and fatigue resistance of these instruments, enabling safer and more efficient shaping of curved and anatomically complex root canals [4].

Concurrently, advances in plasma engineering—particularly cold electric discharge plasmas—have opened new avenues in biomedical fields by enabling targeted microbial inactivation, surface modification, and biological stimulation without thermal damage [5,6,7]. These technologies, which have been widely studied in oncology, pharmacology, and tissue repair, are now being explored for dental applications [8,9]. Despite promising in vitro results, clinical application of cold plasma systems in endodontics remains limited [10,11].

However, prior studies have shown that 35–40% of the canal surface remains untouched during instrumentation due to anatomical complexities such as isthmuses, bifurcations, and lateral canals [12]. Such inaccessible areas can harbor persistent bacteria and biofilms, which may lead to reinfection and endodontic failure, sometimes necessitating retreatment or surgical intervention [13,14].

NaOCl is widely used as an irrigant due to its potent antimicrobial and tissue-dissolving capabilities. However, its use is associated with certain complications. When extruded beyond the root apex, NaOCl can cause severe pain and soft tissue injuries [15,16]. Higher concentrations of NaOCl improve its antimicrobial efficacy but are associated with increased cytotoxicity and a weakening of the root dentin structure over time [17].

In recent years, non-thermal plasma (NTP) technologies have emerged as promising alternatives for microbial control in endodontics. Among these, underwater discharge plasma (UDP) offers a unique approach by generating plasma directly within an aqueous medium [18,19]. This facilitates simultaneous physical and chemical cleaning mechanisms, leveraging reactive oxygen and nitrogen species—such as hydroxyl radicals (^•^OH), hydrogen peroxide, and ozone—that can disrupt bacterial cell membranes, denature proteins, and degrade biofilms [10]. These effects occur at low temperatures (<40 °C), minimizing the risk of thermal injury to periapical tissues, and offer enhanced penetration into anatomically complex regions that are inaccessible to conventional irrigants, all while demonstrating a lower cytotoxic potential compared to high-concentration NaOCl [10]. In vitro studies have shown that UDP exhibits antimicrobial properties against various pathogenic microorganisms, including Escherichia coli and Vibrio cholerae [20,21].

Nevertheless, to date, no prospective clinical studies have evaluated the efficacy or safety of UDP in endodontic therapy in humans. Therefore, in this study, we aimed to clinically assess the effectiveness and safety of UDP in comparison with conventional NaOCl irrigation. This pilot randomized clinical trial evaluated postoperative pain and periapical healing using standardized evaluation methods over a 4-month follow-up period.

## 2. Materials and Methods

### 2.1. Study Design and Ethical Approval

This pilot randomized controlled clinical trial was conducted in accordance with the Declaration of Helsinki and approved by the Institutional Review Board of Hallym University Sacred Heart Hospital (IRB No. 2024-03-016). All patients provided written informed consent prior to enrollment.

### 2.2. Participants

A total of 30 adult patients who visited the Department of Dentistry at Hallym University Sacred Heart Hospital between August 2024 and December 2024 and required RCT were recruited for this study. The inclusion criteria included age between 19 and 90 years and restorable teeth indicated for RCT. Patients were excluded if they had systemic conditions that could affect healing (e.g., uncontrolled diabetes or immunocompromised status), were pregnant, or had a pacemaker or cochlear implant. Additional exclusion criteria included the presence of moderate to severe periodontal disease or an inability to achieve proper rubber dam isolation.

### 2.3. Randomization and Allocation

Patients were randomly assigned in a 1:1 ratio to the UDP (test group, n = 15) or NaOCl (control group, n = 15) groups. The randomization sequence was generated by a third party not involved in treatment or evaluation, and patients were assigned unique study numbers based on the order of enrollment. Two patients (one from each group) were lost to follow-up, resulting in 28 patients completing the study.

### 2.4. Endodontic Procedures

All treatments were performed by a single, experienced dental surgeon, following a standardized protocol. After administering local anesthesia with 2% lidocaine containing 1:100,000 epinephrine, the operative field was isolated using a rubber dam. Access cavity preparation was conducted, and the working length was determined using a K-file and an electronic apex locator (RAYPEX^®^ 6; VDW, Munich, Germany).

In the UDP group, a lip holder from the UDP system (PLAZEN RCT^®^; Dentory, Seoul, Republic of Korea) was first positioned at the corner of the patient’s mouth to complete the electrical circuit required for plasma activation. This configuration enabled the impedance feedback system to monitor and regulate energy delivery throughout the procedure accurately. The root canal was then filled with saline, and a K-file—1 to 2 sizes larger than the initial file—was inserted to a depth 3 mm short of the working length. Plasma discharge was induced by contacting the inserted file with the metal tip of the UDP device, thereby generating high-frequency plasma within the canal. Canal shaping was performed under continuous saline irrigation using VDW.ROTATE™ NiTi rotary files (VDW, Munich, Germany), following the manufacturer’s protocol—typically up to size 25/0.06. In wider canals, instrumentation was selectively extended to size 30/0.05 or 35/0.04. A second plasma discharge was applied immediately prior to obturation.

In the NaOCl group, root canal disinfection was achieved using 6% sodium hypochlorite (Sense Cleaner; Sunjin Bio, Seoul, Republic of Korea). Canal shaping was performed using the same rotary instrumentation protocol as that employed in the UDP group.

In both groups, canals were dried with sterile paper points and obturated using a mineral trioxide aggregate (MTA)-based sealer (Ceraseal^®^; Meta Biomed, Cheongju, Republic of Korea). For cases exhibiting clinical signs of persistent infection or extensive periapical lesions, calcium hydroxide paste (Calcipex II^®^; Nishika, Yamaguchi, Japan) was applied as an intracanal medicament, and obturation was completed at a subsequent visit. Final restorations were placed 1 week after obturation using a resin composite material.

Postoperative pain and periapical healing, assessed using the Periapical Index (PAI) score, were the primary clinical and radiographic outcome measures evaluated in this study. These parameters are widely recognized as core indicators of endodontic treatment success in both clinical and research contexts. According to a recent review, the absence of pain and a PAI score of 1–2 are considered reliable criteria for successful root canal therapy [22]. This outcome assessment framework is also endorsed by the European Society of Endodontology (ESE) S3-level clinical practice guidelines, which designate pain and PAI scoring as principal parameters for evaluating post-treatment outcomes [23].

### 2.5. Pain Assessment

Pain was assessed using a 10-point visual analog scale (VAS), with 0 indicating “no pain” and 10 indicating “worst imaginable pain.” Assessments were conducted preoperatively, 1 week postoperatively, and at 2 and 4 months postoperatively.

### 2.6. Radiographic Assessment

Periapical radiographs were obtained at baseline (preoperative), immediately after canal filling, and at the 2-month and 4-month follow-ups. All images were acquired using the same digital sensor (RIOS Sensor; Healdens Co., Yongin, Republic of Korea) and intraoral X-ray unit (DVAS; Genoray Co., Seongnam, Republic of Korea).

Radiographs were independently evaluated by two board-certified oral and maxillofacial radiologists who were blinded to group allocation. The Periapical Index (PAI) scoring system was used to assess periapical healing [24]. Additional radiographic examples were provided to the reader to assist in interpreting the radiographic images of the teeth (Figure 1).

For multi-rooted teeth, the highest score among the roots was selected as the final score, with a conservative interpretation approach. Each radiologist evaluated the images twice, with a 2-week interval between evaluations, and the median value was used in cases of any discrepancy. Healing was defined as a PAI score of ≤2 and non-healing as a score of ≥3 [24].

### 2.7. Statistical Analysis

The sample size was estimated using G*Power software (ver. 3.1.9.4), and 30 patients (15 per group) were included, achieving a power of 0.65, an effect size of 0.9, and an α level of 0.05 [17]. The normality of continuous variables (VAS and PAI scores) was assessed using the Kolmogorov–Smirnov test. Since the data were non-normally distributed, non-parametric tests were used. The Friedman test was applied to assess within-group changes in VAS scores over time. Intergroup comparisons of VAS and PAI scores at each time point were performed using the Mann–Whitney U test. Differences in healing rates were assessed using Fisher’s exact test. Inter-rater reliability for radiographic assessment was evaluated using the quadratic-weighted Cohen’s kappa coefficient. All tests were two-tailed, and values of *p* < 0.05 were considered statistically significant.

## 3. Results

A total of 28 patients completed the study (14 in each group). No statistically significant difference in baseline characteristics, including age, sex distribution, tooth location (maxilla vs. mandible), and number of canals, was observed between the UDP and NaOCl groups (Table 1).

### 3.1. Pain Assessment Results

Both groups exhibited significant reductions in visual analog scale (VAS) scores over time (UDP group: χ^2^ = 11.28, *p* = 0.010; NaOCl group: χ^2^ = 10.17, *p* = 0.017; Table 2). Intergroup comparisons at each time point (preoperative, 1-week post-filling, 2 months, and 4 months) revealed no statistically significant differences (all *p* > 0.05). At 4 months, the UDP group demonstrated a significantly lower VAS score compared to the NaOCl group (*p* = 0.000; Table 3).

### 3.2. Radiographic Outcomes

Periapical Index (PAI) scores decreased over time in both groups. The mean PAI scores at baseline and follow-up time points (immediate post-filling, 2 months, and 4 months) showed no significant differences between the groups at any time point (all *p* > 0.05; Table 4; Figure 2).

### 3.3. Clinical Success Rates

The rate of clinical success, defined as a PAI score ≤ 2, improved over time in both groups. Success rates at 4 months were 71.4% (10/14) in the UDP group and 92.9% (13/14) in the NaOCl group; however, the difference was not statistically significant (*p* = 0.326; Table 5). The overall healing rates based on radiographic criteria were 82.1% for all patients (Table 6).

### 3.4. Interobserver Agreement

The inter-rater reliability for radiographic PAI scoring was high, with a quadratic-weighted Cohen’s kappa coefficient of 0.721, indicating substantial agreement between the two blinded evaluators.

## 4. Discussion

In this pilot randomized clinical trial, we investigated the clinical effectiveness and safety of UDP as a root canal irrigant compared to NaOCl. Over a 4-month follow-up period, both groups demonstrated significant reductions in postoperative pain and improvements in periapical healing, with no statistically significant differences between them.

Non-thermal atmospheric pressure plasma (NTPP) has been shown to exert strong antimicrobial effects against persistent endodontic pathogens, such as Enterococcus faecalis and Candida albicans, highlighting its potential to overcome the limitations of conventional chemical disinfectants [10]. However, most existing studies are limited to preclinical settings, and clinical validation remains insufficient [10,25].

Akiyama et al. reported that pulsed discharge plasma produces reactive oxygen species (ROS) through physicochemical interactions in water, with antimicrobial efficacy and tissue reactivity strongly influenced by plasma generation conditions and discharge parameters [19]. These findings guided the development of the UDP system, which leverages the moist, enclosed environment of the root canal to facilitate consistent plasma discharge.

UDP can be generated through various mechanisms. One method involves applying high-voltage discharge that ionizes water molecules, forming microbubbles between electrodes. These bubbles act as dielectric barriers, within which subsequent electrical discharges occur. Another method involves heating the electrode surface to its boiling point, creating a vapor layer that initiates ion discharge.

UDP generates plasma through high-frequency, high-voltage electrical discharge in an aqueous environment. When high voltage is applied to a metal tip immersed in fluid, electrons emitted from the tip ionize surrounding water molecules into hydrogen ions (H^+^) and hydroxyl radicals (^•^OH). As more electrons are accelerated through repeated collisions with water molecules, insulation breakdown occurs, creating filamentary discharge and cavitation—the localized formation and collapse of microbubbles. This phenomenon induces sharp increases in temperature and pressure, generating ROS such as ^•^OH, hydrogen peroxide (H_2_O_2_), and ozone (O_3_). Among these, ^•^OH exhibit strong antimicrobial effects by oxidatively damaging bacterial membranes and degrading organic tissue. In addition, the heat and mechanical shockwaves generated by cavitation contribute to a destructive cleansing effect, thermally degrading organic matter and separating it from the internal surface of the root canal system [19,20,26].

Plasma discharge can produce transient, localized heat, which may cause thermal injury to periapical tissues if not adequately controlled. To mitigate the risk of thermal injury caused by transient heat during discharge, the UDP device (PLAZEN RCT^®^; Dentory, Seoul, Republic of Korea) used in this study was designed in accordance with international safety standards (IEC 60601-2-2, KS C IEC 60601-2-2) and incorporates a real-time impedance feedback system. The system continuously monitors the electrical resistance between the tip and the canal contents and automatically stops power delivery when the impedance exceeds 5000 Ω, which typically indicates low conductivity or minimal organic residue [27] (Figure 3).

The correlation between impedance and organic content allows for indirect assessment of canal cleanliness while also guiding safe energy delivery. Based on prior studies, discharge temperatures in the range of 37 °C to 45 °C are generally sufficient for tissue separation without risking overheating [28,29,30]. The device limits each discharge to 1.5 s, and this safety mechanism is indicated on the user interface (Figure 4). Thermal thresholds established by Kim et al. [31] informed the upper temperature limits in the device design.

To verify the generation of ^•^OH—a key component of the antimicrobial mechanism of UDP—a terephthalic acid (TA) fluorescence assay was conducted [32]. In this assay, TA reacts specifically with ^•^OH to produce 2-hydroxyterephthalic acid, which emits fluorescence at 425 nm when excited at 310 nm. A distinct signal was observed spectroscopically and visually (Figure 5), confirming ^•^OH production. Despite their short lifespan (~10^−9^ s), ^•^OH exhibit potent oxidizing capacity, enabling disinfection of areas such as isthmuses and lateral canals that are inaccessible to NaOCl [33,34] (Figure 6).

This study is the first in vivo clinical trial to compare UDP with conventional NaOCl irrigation. At 4 months, the NaOCl group demonstrated a higher radiographic healing rate (92.9%) than the UDP group (71.4%); however, this difference was not statistically significant (*p* > 0.05). PAI score reductions were observed in over 70% of patients in both groups, supporting the potential effectiveness of UDP.

The findings observed in this clinical setting align partially with those of previous in vitro and ex vivo studies investigating non-thermal plasma disinfection in endodontics. Muniz et al. reviewed 17 studies published between 2007 and 2022, noting that direct plasma exposure using helium, argon, or their mixtures showed significant microbial reductions, especially when exposure exceeded 8 minutes [10]. However, many of these studies employed simplified models involving single-rooted or bovine teeth, which limits their clinical applicability. In contrast, the present trial was conducted in vivo under realistic clinical conditions, contributing preliminary evidence of patient-centered outcomes such as pain relief and radiographic healing.

Recent studies have also explored the use of cold atmospheric pressure plasma (CAPP) and plasma-activated liquids (PALs). Banaszak et al. reported a reduction of over 99% in *E. faecalis* with CAPP and PALs within 2–10 min, demonstrating efficacy comparable to or exceeding that of NaOCl and chlorhexidine (CHX), and with fewer adverse effects on dentin microhardness [35]. El Shishiny et al. reported that NTPP achieved a 99.79% reduction in bacterial CFUs within 1 min, outperforming CHX and diode laser treatments [36].

Notably, most existing studies applied plasma in the gas phase via external jets. By contrast, the UDP system generates plasma directly within the irrigating liquid inside the canal, potentially improving energy transfer and minimizing risks such as dehydration or gas expansion [9,10,11]. Whether this liquid-phase advantage translates into superior microbial efficacy remains to be confirmed. Muniz et al. also emphasized the benefit of combining NTPP with chemical irrigants (e.g., NaOCl or CHX), which produced consistently greater antimicrobial outcomes than either alone [10]. Such combination strategies, including sequential use of NaOCl and UDP, were not evaluated in this trial but warrant future investigation.

This study employed UDP with saline. Future trials should include groups with combination protocols (e.g., UDP with NaOCl or CHX) to assess potential additive or synergistic effects. Such studies will help establish optimized irrigation protocols tailored to clinical needs.

Furthermore, standardization of plasma application parameters remains an unmet need. Key variables such as working gas type, exposure time, power setting, and delivery distance influence antimicrobial effectiveness. While exposure times exceeding 10 minutes are generally needed for thorough biofilm eradication in NTPP models [10], UDP application in this study was brief and integrated into canal instrumentation. Optimizing exposure time and total energy delivery may enhance clinical outcomes.

Although intracanal medicament (Calcipex II^®^) was not applied according to a rigid protocol in this pilot trial, its use was limited to cases with persistent exudate or notable symptoms following instrumentation, reflecting real-world clinical decision-making. The broad inclusion criteria—based on the clinical indication for RCT rather than strict diagnostic subtypes—were intended to enhance external validity. Postoperative pain and periapical healing, assessed with the PAI, were chosen as outcome measures in accordance with the current European Society of Endodontology guidelines [23] and recent literature [22], in which these parameters are regarded as core indicators of endodontic treatment success. We acknowledge that these methodological decisions may introduce some variability; however, this approach was deemed appropriate for an initial investigation and will inform the design of more rigorous controlled trials.

As a pilot study, this investigation was exploratory. The sample size was calculated based on an effect size of 0.9 and a power of 0.65, which is below the commonly recommended minimum of 0.8. Although acceptable for hypothesis generation, the limited statistical power restricts the ability to draw definitive conclusions. The 4-month follow-up period may also be insufficient to assess long-term reinfection rates. Nevertheless, early clinical indicators, such as pain reduction and radiographic healing, are valuable outcomes during the initial 3–6 months post-treatment [24,37,38].

The absence of bacteriologic and histologic analyses in this study limits objective evaluation of the antimicrobial mechanism and tissue response. Future studies should include temperature-controlled UDP systems, microbiological validation, and histological assessment to confirm biocompatibility. Comparative studies involving PALs, which maintain activity during storage and may enhance penetration into complex anatomies, are also warranted.

In conclusion, to validate and expand the clinical use of UDP irrigation, future research should incorporate larger sample sizes with adequate power (≥0.8), extended follow-up durations (≥12 months), and multifaceted outcome assessments. Evaluating UDP in conjunction with traditional irrigants such as NaOCl will also clarify whether combination approaches offer superior clinical benefits.

## 5. Conclusions

This randomized pilot study demonstrated that UDP irrigation achieved clinical outcomes comparable to conventional NaOCl irrigation in an RCT. Both groups showed significant reductions in postoperative pain and progressive radiographic signs of periapical healing over the 4-month follow-up, and no adverse reactions or thermal damage were observed with UDP. Although the NaOCl group exhibited a numerically higher healing rate, the difference was not statistically significant, indicating that UDP can support favorable periapical repair within the limitations of this pilot study. These findings suggest that UDP represents a safe and biocompatible alternative to traditional chemical irrigants, with the added potential to reduce risks associated with NaOCl extrusion. As the first in vivo clinical evaluation of this approach, the study provides preliminary evidence for the clinical feasibility of plasma-based irrigation. Nevertheless, larger multicenter studies with extended follow-up are necessary to confirm its long-term efficacy and to optimize application protocols, including potential combination strategies with conventional irrigants.

## Figures and Tables

**Figure 1 biomedicines-13-02343-f001:**
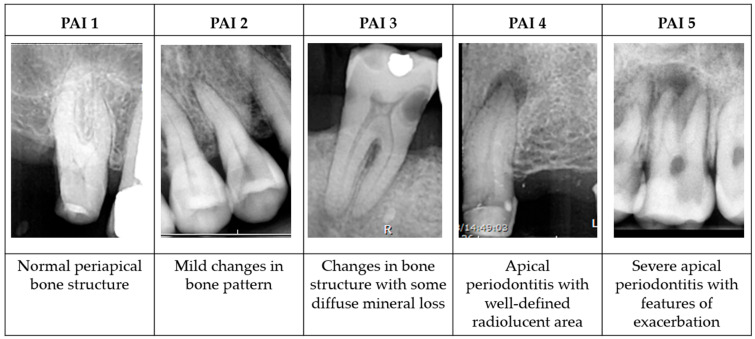
Representative radiographic images with corresponding Periapical Index (PAI) scores. The images depict various degrees of periapical radiolucency, assessed according to the PAI scoring system, introduced by Ørstavik et al. [24], ranging from a healthy periapical status (score 1) to severe periodontitis with exacerbating features (score 5). These reference images were used to assist the blinded radiologists in evaluating periapical healing in this study.

**Figure 2 biomedicines-13-02343-f002:**
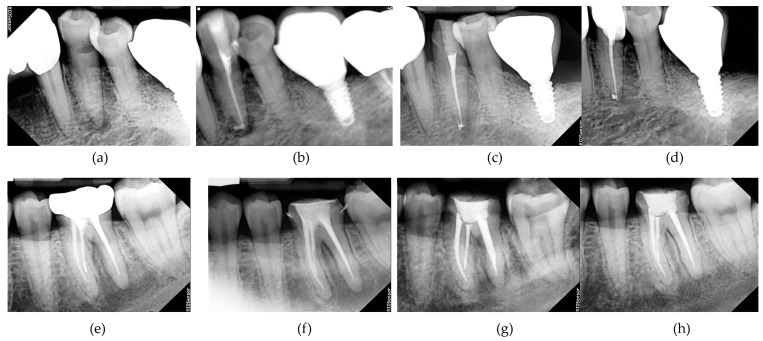
Serial periapical radiographs illustrating the progression of periapical healing in representative cases from the UDP (test) and NaOCl (control) groups. (**a**–**d**) Images from a UDP-treated tooth showing: (**a**) preoperative periapical radiolucency, (**b**) immediate post-obturation status, (**c**) 2-month follow-up status, and (**d**) 4-month follow-up status with evidence of healing. (**e**–**h**) Images from a NaOCl-treated tooth at similar time points. Radiographs were acquired using a standardized digital system (RIOS Sensor; Healdens, Yongin, Republic of Korea) and were used for evaluating PAI scores by blinded evaluators.

**Figure 3 biomedicines-13-02343-f003:**
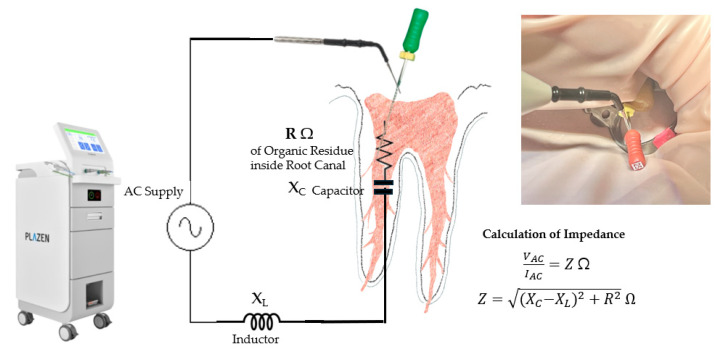
Schematic of the impedance monitoring mechanism in the UDP system. The circuit diagram illustrates how an alternating voltage ($V_{AC}$) is applied and the resulting current ($I_{AC}$) is measured to calculate the total impedance (Z) of the root canal contents. The components include the resistance of organic material (R), inductive reactance ($X_{L}$), and capacitive reactance ($X_{C}$). The total impedance is expressed as $Z=\sqrt{{{(X}_{C}-X_{L})}^{2}+R^{2}}Ω$. The system uses real-time impedance feedback to adjust energy output dynamically and ensure safety. When the impedance exceeds 5000 Ω, the device automatically ceases energy delivery to prevent thermal injury to surrounding tissues.

**Figure 4 biomedicines-13-02343-f004:**
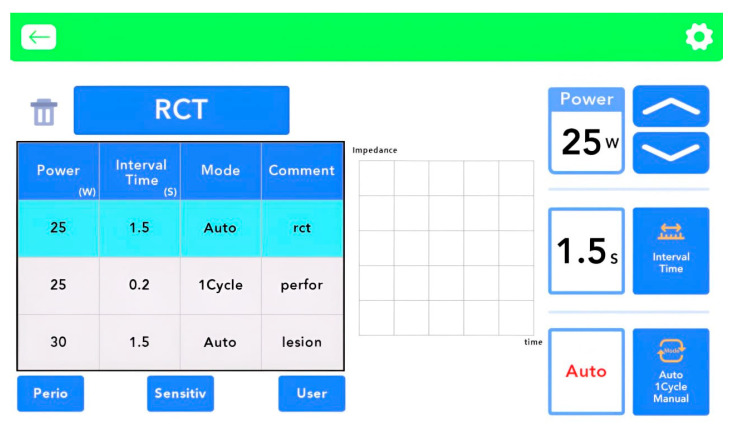
User interface of the UDP device (PLAZEN RCT^®^; Dentory, Seoul, Republic of Korea). The screen displays a preset maximum plasma discharge duration of 1.5 s, a built-in safety feature that regulates energy delivery. This function ensures precise temperature control during plasma activation within the root canal, minimizing the risk of thermal injury.

**Figure 5 biomedicines-13-02343-f005:**
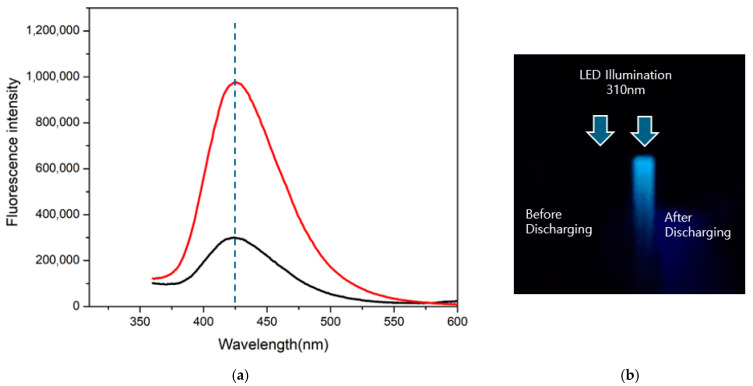
Fluorescence emission analysis of terephthalic acid (TA) solution following underwater discharge plasma (UDP) treatment. (**a**) Emission spectra of the TA solution following pulsed plasma discharge using the PLAZEN RCT^®^ device (Dentory, Seoul, Republic of Korea) (25 W, 1.2 kV, burst wave mode). The black line represents the control sample without plasma exposure, while the red line indicates the plasma-treated sample, which exhibits a distinct emission peak at 425 nm. (**b**) Photographic image of TA solutions under 310 nm LED illumination before (left) and after (right) plasma treatment, captured using the Horiba FluoroMax+ spectrofluorometer (Horiba, Kyoto, Japan) (excitation: 310 nm; emission range: 360–600 nm; increment: 1 nm; slit width: Ex/Em = 3/3 nm).

**Figure 6 biomedicines-13-02343-f006:**
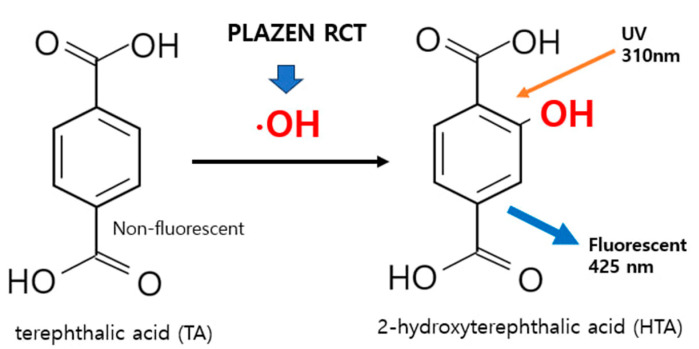
Schematic representation of hydroxyterephthalic acid (HTA) formation through the reaction between terephthalic acid (TA) and hydroxyl radicals (^•^OH). TA reacts with ^•^OH generated by underwater discharge plasma, producing HTA, which emits fluorescence upon UV excitation. This reaction serves as an indirect indicator of the presence of ^•^OH in the solution.

**Table 1 biomedicines-13-02343-t001:** Patient demographics and teeth anatomy.

Group	Men, n (%)	Women, n (%)	Mean Age in Years (mean ± SD)	Single Canal, n (%)	Multiple Canals, n (%)	Maxilla, n (%)	Mandible, n (%)
UDP (n = 14)	7 (50)	7 (50)	50.00 ± 15.48	7 (50)	7 (50)	8 (57)	6 (43)
NaOCl (n = 14)	3 (21)	11 (79)	57.93 ± 14.43	3 (21)	11 (79)	8 (57)	6 (43)
Total (n = 28)	10 (36)	18 (64)	53.96 ± 15.23	10 (36)	18 (64)	16 (57)	12 (43)

NaOCl, sodium hypochlorite; SD, standard deviation; UDP, underwater discharge plasma.

**Table 2 biomedicines-13-02343-t002:** Changes in VAS scores over time within the groups.

Group	Chi-Square (χ^2^)	*p* Value
UDP	11.28	0.010
NaOCl	10.17	0.017

NaOCl, sodium hypochlorite; UDP, underwater discharge plasma; VAS, visual analog scale.

**Table 3 biomedicines-13-02343-t003:** Comparison of VAS scores between the groups at preoperative and follow-up time points.

Time	Mean VAS Score Mean ± SD		*p* Value
	UDP	NaOCl	
Preoperative	1.57 ± 1.99	2.14 ± 2.48	0.603
Postoperative	0.86 ± 1.66	1.14 ± 1.75	0.804
2 months	0.07 ± 0.27	0.36 ± 0.63	0.329
4 months	0.07 ± 0.27	0.14 ± 0.54	0.000

NaOCl, sodium hypochlorite; SD, standard deviation; UDP, underwater discharge plasma; VAS, visual analog scale.

**Table 4 biomedicines-13-02343-t004:** Intergroup comparison of PAI scores at preoperative and follow-up periods.

Time	Mean PAI Score Mean ± SD		*p* Value
	UDP	NaOCl	
Preoperative	2.46 ± 1.24	2.16 ± 1.08	0.603
Canal Filling	2.45 ± 1.24	1.98 ± 0.77	0.376
2 months	1.96 ± 0.88	1.71 ± 0.71	0.511
4 months	1.66 ± 0.79	1.43 ± 0.62	0.401

NaOCl, sodium hypochlorite; PAI, Periapical Index; SD, standard deviation; UDP, underwater discharge plasma; VAS, visual analog scale.

**Table 5 biomedicines-13-02343-t005:** Intergroup comparison of clinical success rates at immediate canal filling and follow-up periods.

Time	Clinical Success Rates (%, n)		*p* Value
	UDP	NaOCl	
CF	50.0 (7)	78.6 (11)	0.236
2 months	57.1 (8)	78.6 (11)	0.420
4 months	71.4 (10)	92.9 (13)	0.326

NaOCl, sodium hypochlorite; UDP, underwater discharge plasma; CF, canal filling.

**Table 6 biomedicines-13-02343-t006:** Comparison of healing rates between the groups at the 4-month follow-up.

Group	Healed, n (%)	Not Healed, n (%)	Improved (Decrease PAI Score)
UDP (n = 14)	10 (71.4)	4 (28.6)	71.4% (n = 10)
NaOCl (n = 14)	13 (92.9)	1 (7.1)	78.6% (n = 11)
Total (n = 28)	23 (82.1)	5 (17.9)	75.0% (n = 21)

NaOCl, sodium hypochlorite; UDP, underwater discharge plasma.

## Data Availability

The data are available from the corresponding author upon reasonable request.

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
