# Peer review of "Clinical Evaluation of Underwater Discharge Plasma as a Root Canal Irrigant: A Randomized Pilot Study on Efficacy and Safety"

_biomedicines, 2025, doi:10.3390/biomedicines13102343_

Round 1
Reviewer 1 Report
Comments and Suggestions for Authors
The aim of the present article was to evaluate the clinical effectiveness and safety of underwater discharge plasma (UDP) as a biocompatible alternative to sodium hypochlorite (NaOCl) for root canal irrigation.
Please specify the studies the authors speak about- Furthermore, while most prior studies employed gas-phase plasma applied through external jets, the UDP system enables plasma formation directly within liquid-filled canals.
Please specify the limitations of the present research.
1. Originality: The article contains new and important information adequate to justify its publication.The topic of the present article is a very interesting one.
2. Fit to the scientifical literature: The present paper demonstrates an adequate understanding of the relevant literature.
3. Methodology: The paper's argument was built on an appropriate base of theory, concept. The methods employed are appropriate and the statistics is well designed.
4. Results: The results are presented clearly, concise, and precise.
5. Discussions: The results analysed appropriately and the conclusion is adequately tie together the other elements of the paper.
5. Implications for research, practice and/or society The paper clearly identify the implications for research, practice and also bridge the gap between theory and practice. These implications consistent with the findings and conclusion of the paper.
6. Quality of Communication: The paper clearly present its case, in an appropriate technical language of the field and at the expected knowledge of the journal's readership.
I suggest the authors in the discussion section to include more recent published articles because there are still a lot of papers published in the scientific literature.
Reviewer 2 Report
Comments and Suggestions for Authors
General Comments:
This manuscript presents a pilot randomized clinical trial assessing the clinical efficacy and safety of underwater discharge plasma (UDP) as a novel root canal irrigant compared to conventional sodium hypochlorite (NaOCl). The study is timely, well-organized, and addresses an innovative approach in endodontic disinfection using non-thermal plasma technology. However, several methodological limitations and reporting omissions reduce the strength and generalizability of the conclusions. The manuscript is of interest but requires substantial clarification and justification in multiple areas.
Major Comments
- Sample Size and Statistical Power
The study is based on a small sample (N=28) and reports a power of 0.65, which falls below the commonly accepted threshold of 0.8. While this may be acceptable for a pilot study, the authors should clearly acknowledge this limitation and avoid overinterpretation of non-significant results. - Short Follow-Up Duration
A four-month follow-up is relatively short in the context of endodontic healing, where periapical outcomes and reinfection often require 6–12 months or longer to manifest. The current timeline limits the ability to draw meaningful conclusions regarding the long-term clinical success of UDP irrigation. - Lack of Microbiological or Histological Validation
While the study evaluates clinical and radiographic outcomes, it lacks microbiological confirmation of UDP’s antimicrobial effect. Without bacterial sampling or histological analysis, claims regarding disinfection efficacy remain inferential. This omission is particularly critical given that the innovation centers on microbial inactivation. - Insufficient Characterization of UDP Parameters
The plasma application method, duration, and energy dose are not thoroughly detailed or standardized. Prior literature indicates that exposure time and device settings are crucial determinants of efficacy. A more precise description of UDP application time and conditions is essential for reproducibility and comparison with existing studies. - Omission of Combination Protocol Evaluation
Previous studies have suggested synergistic effects when non-thermal plasma is used in combination with traditional irrigants. The authors did not include or explore such protocols (e.g., UDP + NaOCl), which would have significantly enhanced the clinical relevance and practical utility of their findings.
Minor Comments
- Device Safety Monitoring: Although the authors mention real-time impedance monitoring, it would strengthen the safety claim to include quantitative temperature data during plasma application to rule out subclinical thermal effects.
- Healing Rate Trends: The NaOCl group showed a higher, though not statistically significant, radiographic healing rate (92.9% vs. 71.4%). While this result is not conclusive due to small sample size, the trend should be cautiously discussed rather than dismissed outright.
- Language and Clarity: The manuscript is generally well-written, but a few sections—particularly in the discussion regarding the physics of plasma generation—could benefit from simplification and clearer linkage to clinical implications.
The authors should:
- Expand the limitations section
- Clarify and justify plasma application methodology
- Temper conclusions regarding efficacy
- Discuss future directions in greater depth, including the need for combination protocols and longer follow-up
Reviewer 3 Report
Comments and Suggestions for Authors
Dear authors,
I appreciate your efforts in conducting this clinical study; however, after thorough review, I must reject the submission in its current form. My major concerns center around methodological inconsistencies that critically impact the validity of the findings.
Firstly, the inconsistent use of intracanal medication (Calcipex II®) is a significant variable. Since intracanal medication directly influences postoperative pain and healing, applying it only in selected cases—based on clinical judgment rather than standardized criteria—introduces serious bias and undermines the comparability of outcomes between groups.
Secondly, the parameters used to evaluate the effectiveness of the irrigant—postoperative pain and radiographic healing—are not sufficient or appropriate to support recommendations regarding irrigant selection. The choice of an endodontic irrigant should be based on a comprehensive analysis of its antimicrobial efficacy, biocompatibility, tissue dissolution capacity, and smear layer removal. Pain and healing outcomes, while clinically relevant, are influenced by multiple variables and do not provide a direct measure of the irrigant’s performance.
In addition:
- The authors said: The aim of this study was to clinically assess the effectiveness and safety of UDP in comparison with conventional NaOCl irrigation. This aim is too vague, please be more specific. Evaluating antimicrobial action? Biocompatibility? Smear layer removal? Post-operative pain? Etc.
- What is the hypothesis of this study?
- The inclusion criteria of (patients aged 19 to 90 years with restorable teeth indicated for RCT), what was the diagnosis? Only indicated for RCT is not sufficient. The vitality of the pulp and the diagnosis highly affect the post-operative pain.
- MTwo files were used up to which size?
- The quantity of each irrigant? The time of irrigation? The technique of irrigation?
- Why did you choose NaOCl at 6%?
